# Gender Differences and Cardiometabolic Risk: The Importance of the Risk Factors

**DOI:** 10.3390/ijms24021588

**Published:** 2023-01-13

**Authors:** Antonella Meloni, Christian Cadeddu, Lucia Cugusi, Maria Pia Donataccio, Martino Deidda, Susanna Sciomer, Sabina Gallina, Cristina Vassalle, Federica Moscucci, Giuseppe Mercuro, Silvia Maffei

**Affiliations:** 1Department of Radiology, Fondazione G. Monasterio CNR-Regione Toscana, 56124 Pisa, Italy; 2Department of Medical Sciences and Public Health, University of Cagliari, 09042 Cagliari, Italy; 3Department of Biomedical Sciences, University of Sassari, 07100 Sassari, Italy; 4Cardiology Unit, European Hospital, 00149 Rome, Italy; 5Department of Clinical and Internal Medicine, Anesthesiology and Cardiovascular Sciences, University of Rome “Sapienza”, Policlinico Umberto I, 00185 Roma, Italy; 6Department of Neuroscience, Imaging and Clinical Sciences, University of Chieti-Pescara, 66100 Chieti, Italy; 7Medicina di Laboratorio, Fondazione G. Monasterio CNR-Regione Toscana, 56124 Pisa, Italy; 8Endocrinologia Cardiovascolare Ginecologica ed Osteoporosi, Fondazione G. Monasterio CNR-Regione Toscana, 56124 Pisa, Italy

**Keywords:** metabolic syndrome, gender, cardiovascular disease, risk factors

## Abstract

Metabolic syndrome (Mets) is a clinical condition characterized by a cluster of major risk factors for cardiovascular disease (CVD) and type 2 diabetes: proatherogenic dyslipidemia, elevated blood pressure, dysglycemia, and abdominal obesity. Each risk factor has an independent effect, but, when aggregated, they become synergistic, doubling the risk of developing cardiovascular diseases and causing a 1.5-fold increase in all-cause mortality. We will highlight gender differences in the epidemiology, etiology, pathophysiology, and clinical expression of the aforementioned Mets components. Moreover, we will discuss gender differences in new biochemical markers of metabolic syndrome and cardiovascular risk.

## 1. Introduction

Metabolic syndrome (MetS) is a complex disorder with a high socioeconomic cost that is generally thought to be a consequence of social and environmental changes related to urbanized living conditions, high-caloric food intake, and sedentary lifestyle [1]. It is considered a worldwide epidemic. MetS is defined by a cluster of causally interconnected metabolic and cardiovascular risk factors (CVRF) such as atherogenic dyslipidemia, arterial hypertension, dysregulated glucose homeostasis, and abdominal obesity. Several MetS definitions, differing in their focus and their diagnostic threshold values, have been proposed by different international organizations, such as the World Health Organization (WHO) [2], the European Group for the study of Insulin Resistance (EGIR) [3], the National Cholesterol Education Programme Adult Treatment Panel III (NCEP ATP III) [4], the American Association of Clinical Endocrinologists (AACE) [5], the International Diabetes Federation (IDF) [6], and the American Heart Association/National Heart, Lung, and Blood Institute [7] (Table 1). Recently, other abnormalities such as chronic proinflammatory and prothrombotic states (characterized by high levels of circulating inflammatory markers such as C-reactive protein and fibrinogen), non-alcoholic fatty liver disease (NAFLD), and sleep apnea have been added as factors of the syndrome, making its definition even more complex [8].

Worldwide, the prevalence of MetS ranges from <10% to as high as 84%, depending on the region, environment (urban or rural), composition (sex, age, race, and ethnicity) of the population studied, and the criteria used [9]. Published reports from different countries differ in the gender distribution of metabolic syndrome: some researchers report a higher incidence in men than women, and the reverse is the case in some other reports. MetS prevalence increases with age [10,11], and different studies have shown a steeper age-related increase in MetS prevalence in women compared with men [10,12] due to a dramatic increase in blood pressure in women after menopause and to a rapid impairment in endothelial function [13].

MetS confers a 5-fold increase in the risk of type 2 diabetes mellitus (T2DM) [14], and a meta-analysis including 87 studies showed that MetS is associated with a 2-fold increase in risk for cardiovascular disease (CVD), myocardial infarction (MI), stroke, and cardiovascular mortality, as well as with a 1.5-fold increase in risk for all-cause mortality [15]. In the RIVANA (Vascular Risk Study in Navarre) Study, including a Mediterranean cohort of 3,976 middle-aged adults, MetS was found to be independently associated with the incidence of CVD, mortality from CVD, and all-cause mortality after adjustment for multiple potential confounders [16]. Indeed, MetS includes both metabolic risk factors promoting atherosclerosis and coronary heart disease (CHD) and hemodynamic risk factors promoting arteriosclerosis and stroke [14]. Moreover, in women, MetS has a higher level of prognostic significance for cardiovascular disease and mortality [17,18], a finding that is consistent with the greater impact of the components of the syndrome in women than in men.

This review will present the current evidence for the differences between genders observed in the epidemiology, etiology, pathophysiology, clinical expression, and management of MetS and its components. Moreover, the ways in which gender differences impact the cardiovascular risk, including pathogenesis, progression, and severity, will be discussed. Finally, factors unique to women that can impact the prevalence and characteristics of MetS in women will be presented.

## 2. Gender Differences in Metabolic Syndrome Components

The individual components that define metabolic syndrome are the same in women and in men, but there are gender differences in how and when these components manifest as well as how they impact the cardiovascular risk in women.

### 2.1. Proatehrogenic Dyslipidemia

Atherogenic dyslipidemia has a direct correlation with CVD. It is a clinical condition characterized by elevated levels of serum triglycerides and small dense low-density lipoprotein (sdLDL) and by low levels of high-density lipoprotein (HDL) cholesterol. Additional features are elevated levels of triglyceride rich in very low-density lipoproteins (VLDL) and apolipoprotein B (ApoB), as well as reduced levels of small HDL [19,20].

The interplay of lipid metabolism with genes, gender, and environmental factors has been shown to modulate disease susceptibility [21]. In particular, a gender-dependent association has been demonstrated between the polymorphisms associated with lipid metabolism at the perilipin locus and obesity risk, supporting the notion that gender-specific differences in morbidity and mortality may be mediated in part by genetic factors associated with lipid metabolism.

It is well-known that premenopausal women exhibit a better lipid profile compared with men, as shown by lower levels of total cholesterol (TC), LDL, and triglycerides along with higher HDL concentrations, which have been partly linked to the specific action of estrogens [22,23]. Indeed, women commonly show better regulation, transport, and removal of VLDL from vessels than their male counterparts [19,20]. On the other hand, several trials have reported a shift toward an unhealthy atherogenic lipid profile in postmenopausal women, who have the tendency to reach higher levels of TC, LDL cholesterol, triglycerides, and lipoprotein(a), and who tend to have lower HDL levels compared with premenopausal women [23]. These menopause-linked changes in the lipid profile are proatherogenic (increased plasma concentration of TC, LDL, and triglycerides) and procoagulatory (higher levels of lipoprotein(a)), and are strongly connected to the increase of visceral fat mass classically associated with menopause-induced modifications [20].

Recent epidemiological data indicate that while in males, high TC and LDL levels are the most important CVRFs, in women, the most significant CVRFs are increased plasma levels of triglycerides and lipoprotein(a). Triglycerides also represent one of the most important risk factors for T2DM in women, but to a lesser extent in men [19,20].

Abdominal fat accumulation, particularly visceral fat (VF) mass, contributes to worsening the dyslipidemic and hypertensive profile detected in women with impaired glucose tolerance [24]. VF accumulation is generally accompanied by insulin resistance (IR), increased release of free fatty acid by adipose tissue, and secretion of ApoB containing particles by the liver, leading to hyperlipidemia. This cascade ultimately results in a preponderance of sdLDL particles and a reduction in antiatherogenic HDL. A similar pattern emerges with menopause, when LDL composition shifts from a low prevalence of sdLDL particles in premenopausal women to one as high as 30%-49% after menopause. These lipid changes are indicative of increased cardiovascular risk and contribute to the number of women meeting the diagnosis of MetS. Thus, monitoring and controlling waist circumference, a marker of abdominal obesity and VF accumulation, represents a key strategy to counteract the clinical consequences of MetS, especially in postmenopausal women [24].

As regards as the therapeutic strategies, the benefits of statins, ezetimibe, and PCSK9 inhibitor therapy have been demonstrated to be comparable between women and men at similar risk levels [25,26]. Nevertheless, women are less likely than men to be treated with any statin or guideline-recommended statin intensity, as well as to receive ezetimibe, most likely because of a lack of appreciation of CVD risk by clinicians [27]. In addition, due to perceived side effects, women are more likely than men to discontinue statin therapy [28].

### 2.2. Arterial Hypertension

Gender differences in the pathophysiology of arterial hypertension seem to be multifactorial and are still not entirely understood [29]. Some of the current hypotheses include differences in sympathetic activation and arterial stiffness, with a specific role of sex hormones [30]. 

The overactivation of the sympathetic nervous system is not only important in the early stages of the development of hypertension, but it is also associated with several comorbidities commonly associated with hypertension [31]. Importantly, autonomic dysfunction seems to play a more prominent role in female than in male hypertension [32]. Moreover, the age-related increase in sympathetic traffic is higher in women than in men, and it is independent of body mass index and menopausal status [33,34]. Therefore, sympathetic neural mechanisms may play a key role in the marked influence of age on blood pressure and cardiovascular disease in women.

Premenopausal women have a lower risk and incidence rate of hypertension compared with age-matched men, but this advantage gradually disappears after menopause. After 65 years of age, the prevalence of arterial hypertension is higher in women than in men [35].

Androgens and estrogens regulate blood pressure (BP) through the renin-angiotensin system (RAS). RAS is stimulated by androgens, resulting in an increase in BP [36], whereas ovarian hormones have the opposite effect, reducing plasma renin and angiotensin-converting enzyme (ACE) activity [37]. Sex hormones’ effects on the reabsorption of renal sodium and on the vascular resistance could also explain the differences in BP control between men and women [38]. Estrogens seem to maintain normal endothelial function by stimulating the production of nitric oxide (NO), inducing structural and functional beneficial effects on the arterial wall that, in turn, reduce vascular stiffness [37]; moreover, they moderate the effects of the sympathetic nervous system [39,40]. 

Antihypertensive therapy is effective for controlling BP in both sexes, but although women take more antihypertensive medications than men, they are less likely to achieve the recommended treatment goals [35,40].

Hypertensive heart disease (HHD) defines the complex and diverse perturbations of cardiac structure and function occurring secondary to hypertension. HHD is frequently characterized by left ventricle hypertrophy (LVH), left atrial enlargement (LAE), and left ventricular (LV) systolic and diastolic dysfunction. Gender influences the LV response to hypertension, with women more likely to develop concentric LVH and men eccentric LVH [41].

Moreover, arterial hypertension is a powerful risk factor for incident heart failure (HF) [42]. According to the Framingham Heart Study, the hazard ratio for developing HF in hypertensive compared with normotensive subjects was about two-fold in men and three-fold in women [43]. Arterial hypertension has the highest population attributable risk (PAR) of all risk factors: 39% for males and 59% for females. Accordingly, a systematic review showed that eight different studies (two prospective, five retrospective, one cross-sectional) concluded that the etiology of HF was more likely to be hypertension for females than for males [44], but a 10 mm reduction in systolic BP could reduce the incidence of congestive HF by 50% [43]. Since women are particularly at risk for arterial hypertension-associated HF, the role of early detection and management of hypertension is even greater than in men.

### 2.3. Dysglycemia

Abnormal glucose homeostasis is commonly diagnosed by establishing the presence of impaired fasting glucose (IFG) and/or impaired glucose tolerance (IGT); these two pathological conditions are not interchangeable and represent metabolically distinct abnormalities characterized by different pathophysiological pathways. In fact, the physiological regulation of fasting glucose depends mainly on hepatic glucose production and hepatic insulin sensitivity, whereas postprandial glycemia in response to a carbohydrate load is regulated by an adequate response of insulin secretion to promote hepatic and muscle glucose uptake, which is dependent on insulin sensitivity [45]. 

The prevalence of IGT and IFG is different between the sexes. The analyses of the study groups of “Diabetes Epidemiology: Collaborative Analysis of Diagnostic Criteria in Europe/Asia” highlighted that IFG is 1.5–3 times more prevalent in men than in women in nearly all age groups, and is 7–8 times more prevalent in older age groups (50–70 years). On the other hand, IGT prevalence is higher in women, with the exception of those over the age of 60 and 80 years in Asian and European populations, respectively [46].

The reason for these differences in early dysglycemia is unknown, but could involve the effect of gonadal hormones. Estrogen and progestin have a direct effect on pancreatic secretion of insulin and on its peripheral action at the pancreatic, muscle, and adipose tissue levels [47]. Animal studies have indicated that natural and synthetic estrogens increase the insulin response to glucose loading [48]. In fact, in rats, insulin levels are reduced by ovariectomy, while they are normalized by the administration of estrogen and progesterone [49]. Accordingly, premenopausal women have lower IR than men of the same age, and this feature represents a factor for their protection against ischemic heart disease (IHD) [50]. 

Bonnet et al. demonstrated that low plasma sexual hormone-binding globulin levels are associated with the onset of hyperglycemia or diabetes only in women, and this finding seems to be partially independent of insulin and adiponectin concentrations [51]. On the other hand, Wang et al. reported that both progesterone and estradiol levels were significantly decreased in IGT and diabetic patients [52].

In addition, the prevalence of T2DM is characterized by a gender difference, being higher in men [53,54]. Indeed, the male gender is usually considered a risk factor for the development of T2DM. However, due to the greater number of elderly women than men and the association between aging and T2DM, there are more women with diabetes than men [55].

In women, diabetes is associated with a 37% increase in mortality rate due to cardiovascular causes compared to that due to a history of myocardial infarction; on the contrary, in men, the presence of a previous MI is associated with a 43% increase in the cardiovascular death rate compared to that determined by the presence of diabetes [56].

The significant gender differences observed in diabetic patients exist due to different pathophysiological processes in men and women. Differences in body composition, fat deposition, mass and activity of brown adipose tissue, and expression of some fat-related biomarkers clearly contribute to the sex-dimorphic diabetes risk. Moreover, predisposition, development, and clinical presentation of diabetes are affected by genetic effects, epigenetic mechanisms, health behavior, nutritional factors, sedentary lifestyle, and stress in different ways in males and females [57,58]. It is well-known that the onset of T2DM in premenopausal women nullifies the cardiovascular protection due to sexual hormones, as evidenced by the reduced endothelium-dependent vasodilatation reserve, which is still higher than that induced in men [59]. In addition, hyperglycemia reduces the production of NO mediated by estrogens [60].

However, gender-specific mortality proceeds beyond the susceptibility of female endothelia to diabetes or other biological factors, and suggests inequalities in therapeutic treatment, leading to underestimation of the problem and/or poor therapy adherence and tolerance [61].

### 2.4. Obesity and Adiposity

Although obesity is undoubtedly influenced by diet, exercise, and genetics, its pathophysiology extends beyond these factors, and an important role is played by the sympathetic nervous system. In fact, it makes a major contribution to the integrated regulation of food intake, involving satiety signals and energy expenditure. The overactivity of the sympathetic nervous system is not only a hallmark of obesity, but it may also take part in the development of metabolic disturbance and cardiovascular complications in obese subjects [62,63].

Central/abdominal (mainly visceral) adiposity has emerged as a better predictor of cardiometabolic risk than general obesity measured by body mass index (BMI) [64,65,66,67]. Central/abdominal fat deposition promotes abnormality in fatty acid metabolism and macrophage accumulation with increased expression of pro-inflammatory biological active compounds, especially adipokines, that enhance oxidative stress (OxS) and endothelial dysfunction. These compounds are also responsible for adipose tissue IR [68]. 

Sex differences in adipose tissue distribution are well-supported by many findings in the literature and are associated with whole-body metabolic health [68]. Premenopausal women tend to accrue more fat in the gluteus–femoral area (lower-body, “ginoid” or “pear” phenotype), predominantly due to a superficial increase in size, and often remain metabolically healthy. Clinical studies conducted in healthy overweight and obese women with a wide range of ages and comorbidities confirm that increased gluteus–femoral fat mass is independently associated with a protective effect on glucose and lipid related cardio-metabolic risk, with a beneficial adipokine profile and fewer pro-inflammatory molecules compared with the subjects with accumulated VF [69,70]. Atherosclerotic protection is also promoted through direct vascular effects; gluteus–femoral fat mass, in fact, is associated with lower aortic calcification and arterial stiffness [71], as well as with a decreased progression of aortic calcification in women [72]. 

The menopausal transition, independently of aging, is associated with adverse changes in body fat distribution, lipid profile, IR, and vascular remodeling. During the perimenopause period, fat deposition shifts to favor the visceral depot that, in addition to the decreased protective effect of estrogens, contributes to endothelial dysfunction, inflammatory state, and arterial stiffness, all of which are markers and causes of female IHD [73,74]. Furthermore, a reduction in adiponectin levels is associated with impaired coronary artery reserve in women with normal epicardial coronary arteries [75,76]. 

Post-menopausal women are also characterized by signs of subclinical atherosclerosis, such as an increased carotid intima–media thickness (IMT) and coronary calcification. The contribution of visceral fat accumulation to subclinical atherosclerosis seems to be higher in females that in males [77]. Inflammation seems to have a possible role in IHD sex differences. Markers of inflammation, such as CRP and IL-6, correlate with measures of adiposity, and this association has been reported to be generally stronger in women than in men, for all measures of adiposity as well as the entity of visceral adiposity [78,79]. 

One aspect of abnormal fat distribution concerns the deposition of fat around blood vessels and the heart. Fat depots around the heart can be classified into pericardial, epicardial, and pericoronary fat. This fat is associated with all features of MetS and with increased insulin resistance [80]. Particularly, perivascular and epicardial fat are also associated with coronary and abdominal aortic calcium, independently from traditional measures of obesity. Although intrathoracic fat is higher in men, the proportion of epicardial/intrathoracic adipose tissue is similar in both genders. The excess cardiovascular adipose tissue appears to be affected by hormonal status in women [81], and its volume is greater after menopause, independent of age, obesity, and other covariates. Moreover, pericardial adipose tissue is associated with coronary artery calcification in women at midlife, playing a possible role in the higher risk of CHD reported after menopause [82]. Excessive fat can also accumulate in cardiomyocytes. Cardiac steatosis could be the link between the observed left ventricular diastolic impairment and IHD in women [83]. In men, the amounts of epicardial and pericardial fat, but not the myocardial fat depots, are independently associated with LV diastolic dysfunction [84]. 

## 3. Impact of Gender on Cardiometabolic Risk in NAFLD

NAFLD is a metabolic disease that is diagnosed when the accumulation of hepatic triglycerides is >5.5% in absence of or with moderate alcohol consumption (i.e., daily intake less than 20 g (2.5 units) in women and less than 30 g (3.75 units) in men) [77]. NAFLD is closely linked with IR and, bidirectionally, with the MetS of which it may be both a cause and a consequence.

Although liver biopsy is still the gold standard for the diagnosis and staging of NAFLD, imaging methods such as ultrasonography, computed tomography, and magnetic resonance imaging have been approved as non-invasive alternative methods. Additionally, raised liver enzymes are used as a surrogate marker [85]. 

Gender and reproductive status modulate the risk of developing NAFLD [86]. Below the age of 50 years, the incidence of NAFLD is higher in the male as compared to the female gender due to the protective effect of estrogens, which wanes after menopause. Accordingly, after the fifth decade of life, postmenopausal women have a similar or even higher prevalence of NAFLD compared to men of the same age. Moreover, women with polycystic ovary syndrome (PCOS) or a history of gestational diabetes mellitus (GDM) have a risk similar to or even higher than that of men.

NAFLD is increasingly recognized as a multisystem disease, and it is associated with increased incidence and prevalence of subclinical and clinical CVD, mainly CHD, independently of age [87]. Moreover, in a study consisting of 132 patients with biopsy-proven NAFLD who were followed for 18 years, CVD was the second-most common cause of death after all of the cancers combined [88]. 

Different studies have focused on the existence of gender differences in the association between NAFLD and CVD/mortality, and many discrepancies are present, mainly due to different populations and diagnosis methods as well as varying definitions of disease. A recent, large cross-sectional study reported that although men had an higher prevalence of ultrasonographic fatty liver disease and carotid plaques, as well as an increased IMT compared to women, ultrasonographic fatty liver disease independently predicted subclinical carotid atherosclerosis (IMT and plaques) only in women [89]. 

One study examining the association between elevated serum alanine aminotransferase (ALT) activity and the 10-year risk of CHD, as estimated using the Framingham risk score, showed that relative increase in risk was much greater in women than in men (hazard ratio 2.14 vs. 1.28) [90] despite the fact that women have a lower absolute risk of CHD than men. However, one transversal study showed that elevated ALT was associated with CHD in men, but not in women [91]. A population-based cohort study from Germany found that increased gamma-glutamyltransferase (GGT) was associated with a higher risk of all-cause and CVD mortality in men, but not in women, and this association was stronger in men who also had ultrasound scanning findings compatible with steatosis [92]. A study based on a national Danish registry showed that, compared to the general population, patients with a hospital diagnosis of fatty liver had a higher all-cause mortality rate, including liver- and CVD-related causes, which was similar between the sexes [93].

## 4. Gender Differences in Biochemical Markers of Cardiometabolic Risk

MetS is characterized by increased concentrations of pro-inflammatory cytokines (Interleukin-6, Tumor Necrosis Factor-α), markers of pro-oxidant status (oxidized LDL, uric acid), prothrombotic factors (Plasminogen Activator Inhibitor-1), and leptin, and by decreased concentrations of anti-inflammatory cytokines (Interleukin-10), ghrelin, adiponectin, and antioxidant factors (paraxonase-1) [94]. Values for many biomarkers differ according to gender; however, whether these differences translate to different impacts on cardiovascular risk and disease remains to be further clarified. 

Interleukin-6 (IL-6) is considered to be one of the cytokines at the top of the inflammatory cascade. Despite some controversial findings, the main body of literature suggests that, compared to men, women have higher IL-6 reactivity to mental and/or physical acute stressors [95,96] and pharmacological inflammatory stimulation [97]. Several reports have described IL-6 as a biomarker in CHD, highlighting a potential point of relevance for IL-6 mediated pathways. A large-cohort prospective study showed that long term IL-6 levels are highly associated with CHD, with the CHD risk increasing continuously with increasing levels of circulating IL-6 concentrations [98]. Another study confirmed a risk association of IL-6 with CHD, including a possible role of IL-6 in mediating the associations of circulating inflammatory markers with the risk of CHD in men [99]. However, no strong evidence of an association between IL-6 and incident CHD was found in older British women after controlling for established CHD risk factors [100]. Further studies need to address whether this could reflect a gender difference.

OxS is a condition that occurs when the rate of reactive oxygen species (ROS) formation exceeds the rate of the antioxidant defense system. Gender is associated with differences in OxS levels and antioxidant enzyme expression, likely related to estrogen antioxidant properties [101]. Accordingly, much data has suggested greater antioxidant potential in females over males, as men appear more susceptible to OxS [101]. In particular, OxS biomarkers are generally found to be higher in men when compared to premenopausal women. However, postmenopausal women show higher levels of OxS biomarkers than men in general populations, as well as in coronary and peripheral artery disease cohorts [102,103]. 

Uric acid (UA) is a commonly used laboratory biomarker, and hyperuricemia is more associated with MetS in females than in males [104]. UA has been primarily identified as a powerful antioxidant; therefore, UA elevation in CVD may represent a compensatory mechanism in response to pro-oxidative and pro-inflammatory status. UA concentration is physiologically lower in women than in men due to the role of steroids in UA regulation, also called “uricosuric effect”, and to the possible urate-depressing effect of estrogens in women. However, emerging findings show that UA is more related with CVD in women than in men [105]. 

Plasminogen Activator Inhibitor-1 (PAI-1) is a critical regulator of the fibrinolytic system. PAI-1 levels are lower in females than in males, likely due to differences in genetics, environmental factors, and/or sex hormones [106,107]. Circulating levels are elevated in patients with CHD and may play an important role in the development of atherothrombosis [108]. In large epidemiological studies, elevated plasma PAI-1 levels have been identified as a predictor of myocardial infarction. No study has assessed the presence of a gender difference in the PAI-1-CVD link. However, it has been demonstrated that the correlations of PAI-1 levels with numerous established CVD-related traits (cholesterol, triglycerides, systolic and diastolic blood pressure, glucose) differ between genders and that the menopausal status strongly affects the patterns of gender differences [106].

Leptin has an important role in the long-term regulation of body weight. It has also been proposed as an independent risk factor for CVD and as an important link between obesity and cardiovascular risk [109]. Plasma leptin levels are higher in women than in men due to the higher proportion of adipose tissue and increased production rate of leptin per unit mass of adipose tissue. A significant association between leptin level and stroke has been demonstrated in women, but not in men, after adjustment for age, smoking, body mass index, waist circumference, and hypertension [110].

Adiponectin is an adipocyte-derived hormone with anti-atherogenic, antidiabetic, and anti-inflammatory properties. Women have higher levels of adiponectin than men and, in addition, postmenopausal women have significantly higher levels of plasma adiponectin than premenopausal women. Clinical studies have implicated hypoadiponectinemia in the pathogenesis of T2DM, coronary artery disease (CAD), and left ventricular hypertrophy. The data in the Framingham Offspring Study indicate that low adiponectin is a significant independent CHD risk factor only in men [111].

Similarly to adiponectin, resistin is another cytokine produced mainly by adipose tissue. Alterations to resistin’s secretion process (increased levels in plasma or expression in metabolic and gonadal tissues) have been observed in some metabolic pathologies (e.g., obesity). Specifically, resistin has been reported to be higher in patients with MetS, and it has been shown to be proportional to increased fat mass, possibly being directly linked to insulin resistance. Thus, it functions as a pro-inflammatory molecule in the presence of obesity and represents a candidate hormone that can potentially link obesity to diabetes [112,113]. Interestingly, this molecule has been identified as one key potential metabolic signal affecting the hypothalamo-pituitary gonadal axis in both sexes [114], while higher levels of resistin have been found in women compared with men, showing different strengths of correlation with cardiometabolic risk factors in the two sexes [115,116]. Another very recent study evidenced that, although blood levels of resistin were higher in women than in men, unstable plaques of men showed significantly greater resistin staining intensity on macrophages/foam cells compared with unstable plaques of women [117]. Thus, whether the local expression of resistin in plaques may better and more directly indicate a possible link between this adipokine and plaque instability than its circulating concentration, sex may modulate this relationship and risk of acute events.

Human serum paraoxonase-1 (PON1) is synthesized in the liver and is physically associated with HDL, on which it is almost exclusively located. It is an antioxidant enzyme and is believed to contribute to the antioxidant and anti-inflammatory properties of HDL. There are gender-related differences in factors independently associated with decreases in PON1, as obesity and obesity-related oxidative stress are more important in females, whereas inflammation is more significant in males. Several articles support the hypothesis that lower serum PON1 activity is related to an increase in plaque formation and, consequently, to a higher risk of CVD [118]. Moreover, low serum PON1 activity is associated with a higher degree of atherosclerosis in patients with confirmed MI or unstable angina pectoris [119]. Lower serum paraoxonase activity has been reported in patients with MI in comparison to the control group [120]. However, it is not known whether this low paraoxonase activity plays a causative role in the pathogenesis of MI or is a consequence of this derangement. In patients with CAD, a significant correlation was found between lower paraoxonase and an increased risk of major adverse cardiac events (death, MI, and stroke), and lower PON activity was observed in men than in women [121].

## 5. Women-Specific Risk Factors for Cardiometabolic Disease

There are unique MetS risk factors in women that act directly or indirectly on CVD risk. These will be discussed below. 

### 5.1. Pregnancy

Pregnancy is a contributor to weight gain and MetS. Normal pregnancy is associated with a shift of coagulation and fibrinolytic systems towards hypercoagulability. Although these changes are aimed at minimizing the risk of blood loss during delivery, they increase the risk of thrombosis three-fold to four-fold. Nulliparous women have lower CVD prevalence compared with parous women (18.0% vs. 30.2%) [122].

Moreover, multiparity is independently associated with higher rates of metabolic syndrome [123]. Women with five or more births have a high (2.27 times) rate of CVD prevalence after adjustment for complications [124].

### 5.2. Gestational Diabetes Mellitus

GDM significantly increases the risk for subsequent glucose intolerance and T2DM (from 2.6% to over 70%) [125,126], as well as for Mets. In fact, Mets is more prevalent in women with a history of GDM compared with healthy controls [127]. The risk is primarily due to increased abdominal obesity. 

Lower HDL, elevated triglyceride levels, and C-reactive proteins are present and significantly enhance the risk of developing CVD [128]. Women with GDM have a higher prevalence of coronary artery disease and/or stroke occurring at a younger age, which is independent of T2DM [129]. Moreover, a recent study demonstrated that GDM is associated with angina pectoris, MI, and hypertension within 7 years postpartum, regardless of subsequent diabetes [130].

### 5.3. Pre-Eclampsia

Pre-eclampsia is defined as a systolic blood pressure of at least 140 mmHg and/or a diastolic blood pressure of at least 90 mmHg on at least two occasions. Proteinuria is present after the 20th week of gestation in women known to be normotensive before pregnancy. Increased pre-pregnancy BMI is a risk factor for pre-eclampsia [131]. Pre-eclampsia increases the risk for subsequent hypertension [132] and diabetes [133,134] in perimenopausal years. The association of both pre-eclampsia and GDM with diabetes and hypertension may arise from common pathogenic pathways. Both conditions are associated with insulin resistance [135,136] and with the presence of endothelial dysfunction and markers of chronic vascular inflammation [137,138]. These entities have been shown to precede the development of overt hyperglycemia in patients at risk for type 2 diabetes [137]. Moreover, defects in insulin sensitivity and secretion are both related to elevated hypertension risk [139]. 

Endothelial damage and increased CVD risk, as well as a higher relative risk of cardiovascular mortality, have been demonstrated in pre-eclampsia [140,141]. According to data from a meta-analysis based on cohort studies, in women with previous pre-eclampsia, the risk of future cardiovascular or cerebrovascular events is doubled compared to unaffected women [142]. Moreover, pre-eclampsia has been demonstrated to remain significantly associated with an increased risk of heart failure, stroke, coronary heart disease, cardiovascular disease, and cardiovascular death after adjusting for potential confounders (age, body mass index, and diabetes mellitus) [143]. Importantly, early-onset pre-eclampsia (before 37 weeks of gestation) conveys a higher CVD risk in comparison with late-onset pre-eclampsia [144]. 

### 5.4. PCOS

PCOS has many characteristics similar to those of the MetS. Women with PCOS show a prevalence of metabolic syndrome of approximately 40% [145]. PCOS and MetS share the same components: central obesity and proatherogenic dyslipidemia. Hypertension, increased fasting glucose levels, and impaired glucose tolerance are also commonly present in PCOS [146]. Moreover, PCOS is an independent risk factor for diabetes, dyslipidemia, obesity, hypertension, and MetS [147]. All of these CVRFs contribute in a synergistic way to endothelial activation, IMT, and preclinical atherosclerosis in younger women [147,148]. Additionally, women with PCOS have a higher risk of hyperfibrinogenemia and thromboembolism than healthy women of similar BMI at all ages [149]. 

Different findings exist regarding the relationship between PCOS and cardiovascular complications and death. Therefore, recently, a meta-analysis comprising 10 cohort studies, for a total of 166,682 women, was performed to better clarify this point. According to the meta-analysis, women with PCOS have an increased risk of cardiovascular and cerebrovascular events, including any CVDs, myocardial infarction, ischemic heart disease, and stroke, but excluding mortality-related outcomes [150].

### 5.5. Menopause

The menopause transition (MT) represents a vulnerable time for women, and its incidental hormonal changes have been associated with unfavorable changes in several indicators of metabolic health, such as negative alterations in the lipid profile, increased susceptibility to weight gain, accumulation of abdominal adiposity, and increased blood glucose [151,152,153]. Therefore, in women, the incidence of MetS and cardiovascular disease increases after menopause, regardless of chronological aging [154,155]. Data also suggest that surgical menopause, with its greater hypoestrogenic effect, increases the risk of MetS by 1.5 times compared to natural menopause [156].

Several MT characteristics have been demonstrated to predict a higher risk of adverse CVD outcomes: early-onset menopause (<45 years of age) [157], higher estradiol levels [158], presence of vasomotor symptoms and other menopausal symptoms [159], and experience of poor sleep or depressive symptoms [160]. 

In light of this evidence, menopause can be viewed as an opportunity for preventative strategies for improved health and longevity in women. Screening for many risks assumes much greater importance after the onset of menopause, thus preventing or attenuating diseases that increase within the first 10 years of menopause [161].

## 6. Conclusions

MetS is a heterogeneous entity, with age and sex variation in component clusters, that may have important implications for interpreting the association between metabolic syndrome and cardiovascular risk during the human lifespan. As in CVD, sex hormones influence gender differences and have a great impact on the expression and outcome of MetS. Thus, the careful identification of gender-specific risk factors is not a mere formality, but a vital cornerstone of full comprehension of cardiovascular and cardiometabolic disease, which are increasingly afflicting the female gender. 

It is evident that there is a need for physicians who approach female patients by stressing the main anamnestic and gender-specific data concerning their hormonal lives, starting from menarche and continuing through pregnancy until menopause. Menopause, in particular, represents a turning moment, and is characterized by hormonal changes that facilitate the onset of a series of diseases such as CHD, stroke, diabetes, osteoporosis, and cognitive decline. 

## Figures and Tables

**Table 1 ijms-24-01588-t001:** Criteria for the diagnosis of metabolic syndrome.

	World Health Organization [2]	European Group for the Study of Insulin Resistance [3]	National Cholesterol Education Programme Adult Treatment Panel III [4]	American Association of Clinical Endocrinologists [5]	International Diabetes Federation [6]	American Heart Association/National Heart, Lung, and Blood Institute [7]
**Criteria**	Insulin resistance + ≥2 other components	Insulin resistance + ≥2 other components	≥3 components	No specified number of factors for diagnosis, left to clinical judgment	Increased waist circumference ≥2 other components	≥3 components
**Dysglycemia**	Impaired glucose regulation or diabetes	Impaired fasting glucose or impaired glucose tolerance (diabetes excluded)	Blood glucose ≥ 110 mg/dL (6.1 mmol/L) or previously diagnosed diabetes	Impaired glucose tolerance (but not diabetes)	Fasting plasma glucose >100 mg/dL (5.6 mmol/L) or previously diagnosed diabetes	Fasting plasma glucose >100 mg/dL (5.6 mmol/L) or on drug treatment for elevated glucose
**Raised plasma triglycerides**	≥150 mg/dL (1.69 mmol/L)	≥150 mg/dL (1.69 mmol/L)	≥150 mg/dL (1.69 mmol/L)	≥150 mg/dL (1.69 mmol/L)	≥150 mg/dL (1.69 mmol/L) or on triglycerides treatment	≥150 mg/dL (1.69 mmol/L) or on triglycerides treatment
**Low HDL cholesterol**	<35 mg/dL (0.90 mmol/L) in men and <39 mg/dL (1.01 mmol/L) in women	<39 mg/dL (1.01 mmol/L) in men and women	<40 mg/dL (1.03 mmol/L) in men and <50 mg/dL (1.29 mmol/L) in women	<40 mg/dL (1.03 mmol/L) in men and <50 mg/dL (1.29 mmol/L) in women	<40 mg/dL (1.03 mmol/L) in men and <50 mg/dL (1.29 mmol/L) in women	<40 mg/dL (1.03 mmol/L) in men and <50 mg/dL (1.29 mmol/L) in women
**Increased blood pressure**	≥160/90 mmHg	≥140/90 mmHg or on antihypertensive medications	≥130/85 mmHg or on antihypertensive medications	≥130/85 mm Hg	≥130/85 mmHg or on antihypertensive medications	≥130/85 mmHg or on antihypertensive medications
**Central obesity**	Waist to hip ratio >0.9 in men and >0.85 in women and/or body mass index >30 kg/m^2^	Waist circumference ≥94 cm in men and ≥80 cm in women	Waist circumference ≥102 cm in men and ≥88 cm in women	Body mass index ≥25 kg/m^2^	Waist circumference > ethnicity-specific thresholds	Waist circumference ≥102 cm in men and ≥88 cm in women
**Other**	Microalbuminuria					

## Data Availability

Not applicable.

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
