# Peer review of "Gender Differences and Cardiometabolic Risk: The Importance of the Risk Factors"

_ijms, 2023, doi:10.3390/ijms24021588_

Round 1

Reviewer 1 Report

Although plenty of research on metabolic syndrome (Mets) has been carried out over the past two decades and numerous publications are available regarding this topic, the current paper discloses all-round and detailed an essential  aspect: the gender differences and importance of  cardiometabolic risk in context of Mets.

Just couple misspellings in the text:

Line No 75. Should insert letter 'e' (Proatherogenic)

Line No166. Probably should be 'The prevalence of IGT and IFG' (not 'The prevalence of IGT and IGF').

The paper brings versatile and well structured information on the topic.

Author Response

We would like to thank the Reviewer for the encouraging feedback and constructive critique and for the effort regarding this manuscript. We have addressed each of the raised concerns, which have substantially improved the manuscript.

Although plenty of research on metabolic syndrome (Mets) has been carried out over the past two decades and numerous publications are available regarding this topic, the current paper discloses all-round and detailed an essential  aspect: the gender differences and importance of  cardiometabolic risk in context of Mets.

Just couple misspellings in the text:

Line No 75. Should insert letter 'e' (Proatherogenic)

Line No166. Probably should be 'The prevalence of IGT and IFG' (not 'The prevalence of IGT and IGF').

A: We have corrected the typo errors.

The paper brings versatile and well structured information on the topic.

A: We thank the Reviewer for the positive feedback.

Reviewer 2 Report

The review "Gender differences and cardiometabolic risk: the importance of the risk factors" written by Meloni et al., is overall well written. This is a comprehensive review that nicely describes the main components that characterize metabolic syndrome and how this complex disorder differs in women and men. 

1. One major comment is for section 4: Gender differences in biochemical markers of cardiometabolic risk.

Do the authors have any thoughts and possibly can expand on Resistin. Along with IL-6 and TNFaResistin has been reported to be higher in patients with MetS and and it has been shown to be proportional to increased fat mass possibly being directly linked to Insulin Resistance (PMID: 11201732; PMID: 11678824). 

2. The last paragraph in the conclusion section of the Review requires to be re-written since it is not very clear what the authors want to communicate (line 453 to 455).

Other minor edits required for this review.

- line 81: environmental factors

- line 82: between polymorphisms associated with lipid metabolism

- line 148: define FHS

-line 184-185: in and IGT? not very clear

- line 192: define MI

- line 244: sentence not clear: abdominal aortic calcium independent of traditional measures of obesity.

- line 251: remove "the" form cardiomyocytes 

- line 266: remove "the" form menopause

- line 445: add reference to the sentence explaining the link between menopause and the rapid impairment of endothelial function. 

Author Response

We would like to thank the Reviewer for the encouraging feedback and constructive critique and for the effort regarding this manuscript. We have addressed each of the raised concerns, which have substantially improved the manuscript.

The review "Gender differences and cardiometabolic risk: the importance of the risk factors" written by Meloni et al., is overall well written. This is a comprehensive review that nicely describes the main components that characterize metabolic syndrome and how this complex disorder differs in women and men. 

  1. One major comment is for section 4: Gender differences in biochemical markers of cardiometabolic risk.

Do the authors have any thoughts and possibly can expand on Resistin. Along with IL-6 and TNFaResistin has been reported to be higher in patients with MetS and and it has been shown to be proportional to increased fat mass possibly being directly linked to Insulin Resistance (PMID: 11201732; PMID: 11678824). 

A: We thank the Reviewer for raising this important point. The following sentences have been added in the text. “Similarly to adiponectin, resistin is another cytokine produced mainly by the adi-pose tissue. Alterations of resistin's secretion process (increased levels in plasma or expression in metabolic and gonadal tissues) are observed in some metabolic patholo-gies (e.g. obesity). Specifically, resistin has been reported to be higher in patients with MetS and it has been shown to be proportional to increased fat mass, possibly being directly linked to insulin resistance. Thus, it functions as a pro-inflammatory molecule in the presence of obesity and represents a candidate hormone that potentially links obesity to diabetes [92,93]. Interestingly, this molecule has been identified as one key potential metabolic signal affecting hypothalamo-pituitary gonadal axis in both sexes [94], whereas higher levels of resistin have been found in women when compared with men, showing different strengths of correlation with cardiometabolic risk factors in the two sexes [95,96]. Another very recent study evidenced that, although blood levels of resistin were higher in women than in men, unstable plaques of men showed significantly greater resistin staining intensity on macrophages/foam cells compared with unstable plaques of women [97]. Thus, whether local expression of resistin in plaques may better and directly indicate the possible link between this adipokine and plaque instability than its circulating concentration, sex may modulate this relationship and risk of acute events.”.

  1. The last paragraph in the conclusion section of the Review requires to be re-written since it is not very clear what the authors want to communicate (line 453 to 455).

A: We have modified the sentence as follows: “Menopause in particular represents a turning moment, characterized by hormonal changes that facilitate the onset of a series of diseases such as CHD and stroke, diabetes, osteoporosis and cognitive decline.”.

Other minor edits required for this review.

- line 81: environmental factors

A: The text has been corrected.

- line 82: between polymorphisms associated with lipid metabolism

A: We have modified the text as suggested-

- line 148: define FHS

A: We have now clarified that FHS stands for Framingham Heart Study

-line 184-185: in and IGT? not very clear

A: We have now eliminated “and”.

- line 192: define MI

A: MI is myocardial infarction.

- line 244: sentence not clear: abdominal aortic calcium independent of traditional measures of obesity.

The sentence has been modified as follows: “Particularly, perivascular and epicardial fat are also associated with coronary and abdominal aortic calcium, independently from traditional measures of obesity.”.

- line 251: remove "the" form cardiomyocytes 

- line 266: remove "the" form menopause

A: We have done the required eliminations.

- line 445: add reference to the sentence explaining the link between menopause and the rapid impairment of endothelial function. 

A: The required reference has been added.

Reviewer 3 Report

The subject is interesting, although not very new. My main concern regards the structure of the article, which is difficult to follow.  The chapter “Gender differences in cardiometabolic risk factors” refers only to 2 medical conditions related to metabolic syndrome (Cardiovascular & Diabetes). I don’t understand why NAFLD is kept apart, as it is also a disease related to the same cardiometabolic risk factors. My suggestion would be either to integrate the remarks concerning NAFLD in the “Gender differences in cardiometabolic risk factors”.

The majority of physiopathological explanations relate to differences in the hormonal profile. It would be interesting to add differences in the autonomic response, when applicable.

It is also worth mentioning that the manuscript was submitted to the "International Journal of Molecular sciences" and the explanation of the pathological mechanisms are at least as important as the epidemiological data. In many sections of the manuscript the molecular support misses or is too general. Please consider this comment for the revised version.

I have also some more specific comments which are summarized in the attached document.

Author Response

We would like to thank the Reviewer for the encouraging feedback and constructive critique and for the effort regarding this manuscript. We have addressed each of the raised concerns, which have substantially improved the manuscript.

The subject is interesting, although not very new. My main concern regards the structure of the article, which is difficult to follow.  The chapter “Gender differences in cardiometabolic risk factors” refers only to 2 medical conditions related to metabolic syndrome (Cardiovascular & Diabetes). I don’t understand why NAFLD is kept apart, as it is also a disease related to the same cardiometabolic risk factors. My suggestion would be either to integrate the remarks concerning NAFLD in the “Gender differences in cardiometabolic risk factors”.

The majority of physiopathological explanations relate to differences in the hormonal profile. It would be interesting to add differences in the autonomic response, when applicable.

A: We have now changed the title in “Gender difference in metabolic syndrome components”. In fact, our idea was to describe in this chapter the individual components that define the metabolic syndrome. This is why we described the NAFLD in a different chapter.

It is also worth mentioning that the manuscript was submitted to the "International Journal of Molecular sciences" and the explanation of the pathological mechanisms are at least as important as the epidemiological data. In many sections of the manuscript the molecular support misses or is too general. Please consider this comment for the revised version.

A: Thank you for this comment. We have tried to add more information about pathological mechanisms and differences in the autonomic response.

I have also some more specific comments which are summarized in the attached document.

166 I suppose it is a spelling mistake: IGF is, in fact IFG

A: the spelling mistake has been corrected.

  1. The affirmation about the differences in the pathophysiological processes is not enough

supported, as there is no comment on what happens in men.

A: A complete and comprehensive description of the gender differences in risk and pathophysiology of diabetes would be too long. Anyway, we have now briefly commented this point in the text as follows. “Differences in body composition, fat deposition, mass and activity of brown adipose tissue, and expression of some fat-related biomarkers clearly contribute to sex-dimorphic diabetes risk. Moreover, predisposition, development, and clinical presentation of diabetes are affected by genetic effects, epigenetic mechanisms, health behavior, nutritional factors, sedentary lifestyle, and stress in a different way in males and females [53,54].”

229-232 – what is the relation of this phrase with the title “obesity and adipose tissue”? Is any

connection with adiponectin proven? Or is it just a supposed relation speculated by the authors?

A: The sentence has now been eliminated.

253 – the reference [69] indeed considers only women but there are similar findings in men

(Nyman K, Granér M, Pentikäinen MO, Lundbom J, Hakkarainen A, Sirén R, Nieminen MS,

Taskinen MR, Lundbom N, Lauerma K. Cardiac steatosis and left ventricular function in men

with metabolic syndrome. J Cardiovasc Magn Reson. 2013 Nov 14;15(1):103. doi:

10.1186/1532-429X-15-103. ) which does not support the idea of a difference.

A: We have now added the following sentence. “In men the amounts of epicardial and pericardial fat, but not the myocardial fat depots, are independently associated with LV diastolic dysfunction [74].”.

295-367. In the “Gender differences in biochemical markers of cardiometabolic risk” section several biomarkers (IL-6, PAI-1) are presented as having no sex-related differences, although there studies (which need to be revised and critically upraised) about these differences. At least the paragraphs referring to these biomarkers should be revised.

A: We really thank the Reviewer for this comment. The following sentences have been added.

Despite some controversial findings, the main body of the literature suggests that, compared to men, women have higher IL-6 reactivity to mental and/or physical acute stressors [85,86] and pharmacological inflammatory stimulation [87].

PAI-1 levels are lower in females than in males, likely due to differences in genetics, environmental factors, and/or sex hormones [96,97]. …………However, it has been demonstrated that the correlation of PAI-1 levels with numerous established CVD-related traits (cholesterol, triglycerides, systolic and diastolic blood pressure, glucose) differ between genders and that the menopausal status strongly affects the patterns of gender differences [96].

365- the authors state that they will present epidemiological and pathophysiological data in the introduction. In the section “Women-specific risk factors for cardiometabolic disease”, the mechanism by which the pathological pregnancy, gestational diabetes and preeclampsia have a long term influence and increase the risk of hypertension and diabetes, as components of the metabolic syndrome, is not presented.

A: We thank the Reviewer for this comment. We have now added the following sentences. “The association of both preeclampsia and GDM with diabetes and hypertension may arise from common pathogenic pathways. Both conditions are associated with insulin resistance [135,136] and with the presence of endothelial dysfunction and markers of chronic vascular inflammation [137,138]. These entities have been shown to precede the development of overt hyperglycemia in patients at risk for type 2 diabetes [137]. Moreover, defects in insulin sensitivity and secretion are both related to elevated hypertension risk [139].”.

Conclusions should be based on the features already presented in the article, without further references. Conclusions should be just conclusive remarks, they don’t have to add further arguments. The results from the references included in the section “Conclusion” should be used as arguments in the main text of the article

A: We have now moved different parts of the Conclusion in the Introduction.